# Bayesian Importance of Features (BIF)

## Abstract

We introduce a framework that provides quantitative explanations of statistical models through the probabilistic assessment of input feature importance. The core idea comes from utilizing the Dirichlet distribution to define the importance of input features and learning it via approximate Bayesian inference. The learned importance has probabilistic interpretation and provides the relative significance of each input feature to a model's output, additionally assessing confidence about its importance quantification. As a consequence of using the Dirichlet distribution over the explanations, we can define a closed-form divergence to gauge the similarity between learned importance under different models. We use this divergence to study the feature importance explainability tradeoffs with essential notions in modern machine learning, such as privacy and fairness. Furthermore, BIF can work on two levels: *global* explanation (feature importance across all data instances) and *local* explanation (individual feature importance for each data instance). We show the effectiveness of our method on a variety of synthetic and real datasets, taking into account both tabular and image datasets. The code can be found at `https://anonymous.4open.science/r/BIF-45EF/`

## 1 Introduction

The increasing spread of machine learning algorithms in a wide range of human domains prompt the public and regulatory bodies to increase scrutiny on the requirements on algorithmic design (Voigt & Bussche, 2017). Among them, *explainability* aims to provide a human interpretable reasoning for algorithmic decisions (Goodman & Flaxman, 2016). Quantifying explainability is a difficult problem since numbers should describe characteristics which are human interpretable. With the goal of gaining explainability, there has been a large number of methods developed for explaining why machine learning models produce particular outputs (Ribeiro et al., 2016a; Lundberg & Lee, 2017b). However, the interpretation of the score is often not straightforward. We tackle the problem of accurately assessing how relevant each of the data input feature is to a machine learning model. We formulate the feature importance problem in an intuitive probabilistic framework where the score signifies the relative importance of a feature compared to others. Besides, it is often overlooked how explainability scores relate to other desirable properties of models such as privacy or fairness. The proposed probabilistic interpretation in form of Dirichlet distribution allows closed-form KL-divergence to compare probability distributions that describe explainability of different models.

Broadly speaking, there are two levels at which one could gain such input feature explainability. The first level is *global*, where the goal is to identify the most relevant features for the entire dataset, that is each data sample has the same hierarchy of features. However, when the data exhibit a large variability (e.g. in the case of images), using global explanation methods is not suitable, as for each data sample different features may be relevant. To overcome this limitation, the second level is *local*, where the goal is to identify the most relevant features for each data instance separately (Shrikumar et al., 2017). The current literature proposes separate mechanisms for each of the approaches to feature explanations. The traditional feature selection literature offers a large number of methods for global explanations (Hall, 1999; Hanchuan Peng et al., 2005; Candes et al., 2016; Guyon & Elisseeff, 2003; Kira & Rendell, 1992) while newer methods concentrate on instance-wise approaches (Chen et al., 2018; Yoon et al., 2018). On the other hand, BIF is a common framework which allows both for global and local explanation.

Providing local explanations is challenging and requires backing by an external model. The current approaches provide two ways for local explanations. The first group provides model-agnostic methods which rely on the outputs of common machine learning models (e.g. Random Forest or XGBoost). The second group concentrates on leveraging neural networks to build their own models which provide feature explanations. BIF is a novel method that bridges both approaches, additionally providing a Bayesian perspective for feature importance.

Thus, we introduce a framework called *Bayesian importance of features (BIF)* for gaining the explainability of complex machine learning models both globally and locally with the following benefits:

- It models feature importance via Dirichlet distribution thus providing probabilistic interpretation and relative weighing of the features.

- By means of Bayesian formulation, it provides the uncertainty measure of the assessed importance values.

- The algorithm produces the distribution over the feature importance, which we exploit to quantify the trade-offs between a model's explainability in terms of the feature importance and other notions such as privacy and fairness.

- As a unified framework, it can learn a global probability over features via a simple model linear in the number of parameters, or apply neural networks to produce an instance-wise explanation.

- The proposed method is a flexible meta-algorithm which can work with any model through which we can backpropagate.

## 2 Related Work

Feature selection, overall, is intended to reduce the number of data input variables to those that are the most useful to a model in order to predict the target variable. The feature selection methods can be subdivided in various ways, supervised and unsupervised methods, or wrapper (which evaluate subsets of variables to maximize performance), filter (which evaluate the relationships between each input feature and the target) and embedded (such as penalized regression or random forest) methods (Kuhn et al., 2013). Feature importance, the issue which we tackle in this work, belongs to the filter-type methods. Alternatively, we may also distinguish two broad categories, *feature-additive* and *feature-selection* methods (Linardatos et al., 2021).

Feature-additive methods provide importance of features per dimension, such that their sum matches a quantity of interest, typically the model's output. Two popular such methods are LIME (Ribeiro et al., 2016b) and SHAP (Lundberg & Lee, 2017a). LIME assumes that any complex model is linear locally. It fits a simple model around a single observation using new samples with permuted features, weighted according to their proximity to the original. SHAP is based on the game-theoretical concept of the Shapley value which assesses average marginal contribution of an input. SHAP, however, utilizes the reinterpretation from (Charnes et al., 1988) where the score is a weighted linear combination of features. This approach provides a local explanation by the level of deviation a given data sample gets from the global feature average. Both LIME and SHAP provide a feature importance weight. Other notable feature additive approaches include attribution-based methods (Ancona et al., 2017) such as Integrated Gradient (Sundararajan et al., 2017), Smoothgrad (Smilkov et al., 2017).

On the other hand, feature-selection narrows down the set of features and finds a subset of input features which produce a similar result to the case when the full set of input features is used. Two recent such models are L2X (Chen et al., 2018) and INVASE (Yoon et al., 2018). One notable difference from the classical feature selection methods such as LASSO (Tibshirani, 1996) is that these methods target instance-wise feature selection and build their own neural network-based pipeline. L2X maximizes the mutual information between subsets of features $X_s$ and the response variable $y$, and approximates this quantity with a network which produces binary feature samples learnt via continuous relaxation with the Gumbel-softmax trick. INVASE (Yoon et al., 2018) consists of three networks. The first network is a selector network which provides selection

probabilities given each input feature. However, unlike in our model, the outputs of the selector network are treated as separate Bernoulli variables and are not directly trained with the model's output. The output of the selector network is fed into two separate networks which form an actor-critic pair for feature selection. Both of these methods produce binary output of $k$ important features: L2X outputs a predetermined set of important features while INVASE determines the $k$ based on a threshold.

# 3 Problem formulation

Generally, *feature importance* assigns a score $f \in \mathbb{R}$ to each feature in the input data $\mathcal{D} \in \mathbb{R}^{N \times S}$, where $S$ denotes the number of features and $N$ the number of data examples. For a given feature, the score can be the same or differ for each data sample $\mathbf{x} \in \mathcal{D}$, which is described by the two following concepts:

*Global explanation*: We assign a universal, $S$-dimensional vector of values $\mathbf{f}$ to the set of input features in the dataset $\mathcal{D}$. The value quantifies the relevance or importance of the feature for the entire dataset. In particular, the vector $\mathbf{f}$ assigns a single, scalar-valued global score $\mathbf{f}^{[j]}$ to each feature $j$.

*Local explanation*: Every instance of data $\mathbf{x}_n$ is assigned a separate, $S$-dimensional feature importance vector $\mathbf{f}_n$, and the importance vectors $\mathbf{f}_m$ and $\mathbf{f}_n$ for two different data instances $\mathbf{x}_m$ and $\mathbf{x}_n$ need not be the same.

Moreover, feature importance is computed in relation to a model $g$. In other words, a feature is important for a task indicated by a model in question, denoted by $g$, and thus we require a data-model tuple $(\mathcal{D}, g)$. What comes next describes our method that provides probabilistic interpretation of both global and local explanations.

# 4 Methods: Bayesian importance of features (BIF)

Consider a data-model tuple $(\mathcal{D}, g)$ such that the dataset $\mathcal{D} \in \mathbb{R}^{N \times S}$ and $g$ is any differentiable model. That is, a model $g$ is trained with an $N$-element dataset $\mathcal{D} = \{\mathbf{x}_n, y_n\}_{n=1}^N$, where $\mathbf{x}_n \in \mathbb{R}^S$ is an input data sample, $y_n$ its label (either discrete or continuous) and $S$ is the input dimension. The proposed method assesses the importance of the set of $S$ input features given the model $g$. We assume the model $g$ to be fixed, that is the parameters of $g$ are not altered throughout the process of learning the importance of features. Subsequently, we describe in detail how BIF works in both global and local setting.

## 4.1 BIF for global explanation

In the global feature explanations, we assign the importance for a feature across the entire dataset. For example, in the task to predict future stroke, the database contains information about patients. In this case, the heart pressure measurement is relevant across the dataset for all the patients while their income feature may be less relevant. Feature importance assigns a numerical value that represents the importance of each feature. Usually, the importance value is unbounded (Lundberg & Lee, 2017a) or binary. We define the feature importance through a probability vector $\mathbf{f}$, *importance vector*, which describes the relative weight of a feature.

In the case of 0-1 interpretation of feature importance, the features which are deemed important are preserved and those irrelevant are removed. This process can be seen as an element-wise (Hadamard) multiplication of the binary importance vector and the data vector resulting in the new data sample which contains only the relevant features. We follow a similar train of thought, however, in our case the importance weights are continuous values between 0 and 1. The mechanism is illustrated in Fig. 1. The importance vector $\mathbf{f}$ is element-wise multiplied with the input data, $\mathbf{f} \odot \mathbf{x}$ which results in the new weighted input to the model $g$.

The provided interpretation allows for a more fine-grained analysis of the feature importance. Moreover, the importance can be viewed as a contribution of a feature $\mathbf{f}^{[j]}$ to the maximization of the objective function given by the model $g$.

***Loss function for global explanation*** $\mathcal{L}_G$***:*** Let $\mathcal{L}_G$ denote the loss for the method to obtain the global feature importance. In this Bayesian approach, the data $\mathcal{D}$ is assumed to come from a probability distribution,

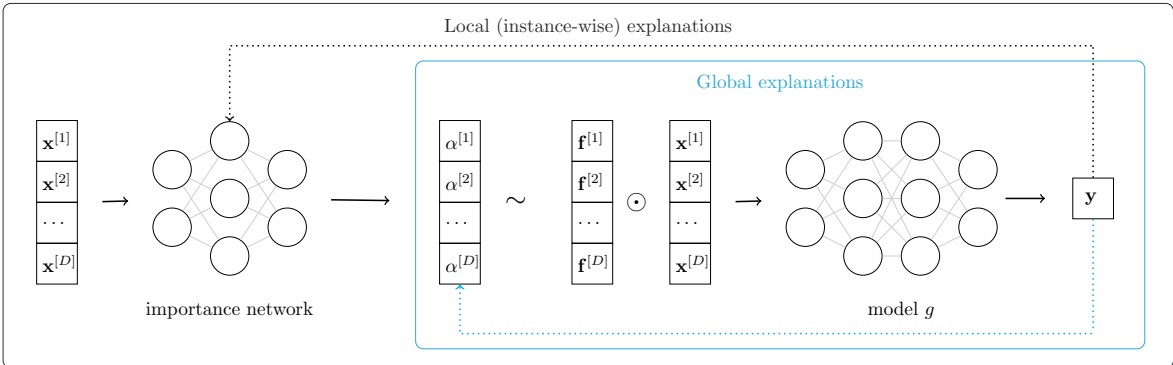

**Figure 1: Global explanation** (blue box). We define an importance vector, assumed to be Dirichlet distributed. To learn the parameters of this distribution (denoted by $\alpha^{[1]} \cdots \alpha^{[S]}$), we element-wise multiply a sample of this distribution ($\mathbf{f}^{[1]}, \cdots, \mathbf{f}^{[S]}$) by input features ($\mathbf{x}^{[1]}, \cdots, \mathbf{x}^{[S]}$), which is fed into the model in question (denoted by $g$). The model's prediction $\mathbf{y}$ and the target determine the loss, used to update the parameters of Dirichlet distribution through back-propagation (blue dotted line). **Local explanation** (black box). To gain instance-wise, local explanation, we define an importance network, where the inputs are the input features ($\mathbf{x}^{[1]}, \cdots, \mathbf{x}^{[S]})$) and outputs follow Dirichlet distributions. We learn the parameters of the importance network via back-propagation (black dotted line). Note that the back-propagation does *not* affect the model in question and that we train either the importance vector or the importance network depending on the application.

and we are interested in inferring this distribution, in particular, the distribution that maximizes the likelihood of data $\mathcal{D}$. We assume there exists a hidden (latent) variable $\mathbf{f}$ and we condition our observations on this variable $\mathbf{f}$. That is, we assume a parametrized model of joint distribution between the data $\mathcal{D}^1$ and the importance feature vector $\mathbf{f}$. In principle, we want to maximize the log-likelihood of the data which is obtained by integrating out the feature importance vector, $\log p(\mathcal{D}) = \log \int p(\mathcal{D}, \mathbf{f}) d\mathbf{f}$. Under a neural network model, directly integrating out $\mathbf{f}$ is intractable, therefore approximate inference is performed. We instead use a distribution $q(\mathbf{f})$, which approximates $p(\mathbf{f}|\mathcal{D})$, $\log \int p(\mathcal{D}, \mathbf{f}) \frac{q(\mathbf{f})}{q(\mathbf{f})} d\mathbf{f}$. By using the Jensen inequality, we can provide the lower bound for this expression in the form of:

$$\int q(\mathbf{f}) \log p(\mathcal{D}|\mathbf{f}) d\mathbf{f} - D_{KL}[q(\mathbf{f})||p(\mathbf{f})]. \tag{1}$$

This expression provides the lower bound to the data likelihood, $p(\mathcal{D})$. The left-hand side is a cross-entropy term obtained from the model $g$ (e.g. neural network output) for a given sample $\mathbf{f}$. The right-hand is the KL divergence between the prior term and the approximate posterior. The negative lower bound becomes our global loss function, $\mathcal{L}_G$ which we minimize.

***Parameterization***. In order to perform approximate inference, we assume the following forms of distribution of the importance vector $\mathbf{f}$. The terms in $\mathcal{L}_G$ are defined as follows:

$$q(\mathbf{f}) = \text{Dir}(\mathbf{f}|\boldsymbol{\alpha}) \quad \text{(approximate posterior)}, \tag{2}$$
$$p(\mathbf{f}) = \text{Dir}(\mathbf{f}|\boldsymbol{\alpha}_0) \quad \text{(prior)} \tag{3}$$

Both $\boldsymbol{\alpha}$ and $\boldsymbol{\alpha}_0$ are parameter vectors of the Dirichlet distribution. We set the parameters $\boldsymbol{\alpha}_0$ to some constant value and only optimize for $\boldsymbol{\alpha}$. Assuming the Dirichlet distribution both for the posterior and the prior allows us to obtain a closed-form KL-divergence in Eq. 1. Thus, the objective function in Eq. 1 depends

---

[1]We preserve the notation $\mathcal{D}$ which is common in the literature, however, one should note that in the derivations $\mathcal{D}$ is an equivalent notation for a sample $\mathbf{x}_n$. For the global case, we omit the index $n$ for clarity.

on the Dirichlet parameters $\boldsymbol{\alpha}$:

$$\mathcal{L}_G(\boldsymbol{\alpha}) := -\int q(\mathbf{f}|\boldsymbol{\alpha}) \log p(\mathcal{D}|\mathbf{f}) d\mathbf{f} + D_{KL}[q(\mathbf{f}|\boldsymbol{\alpha})||p(\mathbf{f}|\boldsymbol{\alpha}_0)]. \tag{4}$$

The crucial characteristic of the global explanation when computing the loss is that for two samples, $\mathbf{x}_m$ and $\mathbf{x}_n$, the importance vectors $\mathbf{f}_m$ and $\mathbf{f}_n$ are sampled from the same parameters $\mathrm{Dir}(\boldsymbol{\alpha})$ across the entire dataset. In the above loss we use the likelihood of the model $g$ (that is $p(\mathcal{D}|\mathbf{f})$ which is computed with the output of the model $g$), but we do not alter the parameters of the model $g$ (which we assume to be pre-trained with $\mathcal{D}$). We freeze its parameters, and only train the parameters of the feature importance vector $\mathbf{f}$. The algorithm for obtaining the global feature importance is summarized in Algorithm 1[2].

***Why Dirichlet?*** Dirichlet distribution describes a family of categorical distributions defined over a simplex, and a sample of the Dirichlet distribution is a probability vector, where all elements of $\mathbf{f}$ are non-negative and $\sum_{i=1}^{S} \mathbf{f}^{[i]} = 1$. This property makes it natural to model a relative level of importance across different input features. Moreover, the choice of Dirichlet distribution allows for the closed-form expression of KL-divergence.

## 4.2 Local or instance-wise explanation via Importance Network (IN)

Local explanations differ from the global ones in that feature importance is evaluated for each data instance. In global explanation, we only produce a single vector $\mathbf{f}$. Conversely, in the local setting, we produce an importance matrix. To be precise, each data point $\mathbf{x}_n \in \mathcal{D}$ is assigned a vector $\mathbf{f}_n \in [0,1]^S$ indicating the *feature importance* for that particular data point. Thus, while $|\mathcal{D}| = N$, an importance matrix of size $N \times S$ is generated. We summarize our algorithm in Algorithm 2 and provide the graphical depiction of this process in Fig. 1 (black box).

***Parameterization.*** In the case of local explanations, each of the importance vectors, $\mathbf{f}_n$ is also modelled by the Dirichlet distribution with individual parameters $\boldsymbol{\alpha}_n$,

$$q(\mathbf{f}_n) = \mathrm{Dir}(\mathbf{f}_n|\boldsymbol{\alpha}_n) \quad \text{(approximate posterior)} \tag{5}$$
$$p(\mathbf{f}_n) = \mathrm{Dir}(\mathbf{f}_n|\boldsymbol{\alpha}_0) \quad \text{(prior)} \tag{6}$$

where we set the parameters $\boldsymbol{\alpha}_0$ the same for all $\mathbf{x}_n$, as we do not have any prior knowledge on any particular data instances.

***Importance network.*** In the local explanations, we assign an importance vector $\mathbf{f}_n$ for each data instance $\mathbf{x}_n$. To learn the mapping between the two, we resort to an additional model, an *importance network (IN)*[3] parameterized by $\boldsymbol{\theta}$. The importance network maps a data instance $\mathbf{x}_n$ to a corresponding Dirichlet parameter vector $\boldsymbol{\alpha}_n$, i.e., $\mathrm{IN}_{\boldsymbol{\theta}} : \mathbf{x}_n \mapsto \boldsymbol{\alpha}_n$. And following Eq. 6, we draw a corresponding feature importance $\mathbf{f}_n$ from the Dirichlet distribution with the parameter $\boldsymbol{\alpha}_n$. Hence, the IN model can produce an individual feature importance for each data instance (via Dirichlet parameters).

***Loss function for local explanation $\mathcal{L}_L$:*** Similarly to the global case, we again use the lower bound derived from Eq. 1 . However, in the local case, our new objective function becomes dependent on the parameters of the importance network:

$$\mathcal{L}_L = -\sum_{n=1}^{N} \int q(\mathbf{f}_n|\boldsymbol{\theta}) \log p(\mathbf{x}_n|\mathbf{f}_n) d\mathbf{f}_n + D_{KL}[q(\mathbf{f}_n|\boldsymbol{\theta})||p(\mathbf{f}_n|\boldsymbol{\alpha}_0)] \tag{7}$$

During the training, we optimize for $\boldsymbol{\theta}$, the parameters of the importance network. Given a sample $\mathbf{f}_n$ (as in $q(\mathbf{f}_n|\boldsymbol{\theta})$), we apply an element-wise multiplication, $\mathbf{f}_n \circ \mathbf{x}_n$, which is fed to the model in question $g$. The model $g$ then produces the conditional distribution and the corresponding loss, $p(\mathbf{x}_n|\mathbf{f}_n)$. During learning as in the global case, we do not update the parameters of $g$, but only update the parameters of the IN model. Besides, $\boldsymbol{\alpha}_0$ is set to a fixed value.

---

[2]Our algorithm is general for any classification (both binary and multi-class) and regression tasks. However, in our experiments we focus on the classification tasks.

[3]In our experiments, we use a multi-layer feed-forward network. Note that other types of networks are also possible, e.g. convolutional neural networks can be more appropriate for image data.

---

**Algorithm 1** Global BIF

---

**Require:** Model in question $g$ with *fixed* weights
1: **for each** train-mini-batch $b$ **do**
2:     Sample $\mathbf{f}$ as in Eq. 3
3:     Compute $g(\mathbf{f} \circ \mathbf{x}_n)$ for $\mathbf{x}_n \in b$
4:     Update $\boldsymbol{\alpha}$ by maximizing $\mathcal{L}_G$ in Eq. 4.
5: **end for**
6: **return** Dirichlet parameters $\boldsymbol{\alpha}$ for global explanation

---

**Algorithm 2** Local BIF

---

**Require:** Model in question $g$ with *fixed* weights
1: **for each** train-mini-batch $b$ **do**
2:     Compute $\boldsymbol{\alpha}_n$ for each $\mathbf{x}_n$ using $\text{IN}_{\boldsymbol{\theta}}$
3:     Given $\boldsymbol{\alpha}_n$, sample $\mathbf{f_n}$ from Eq. 6
4:     Compute $g(\mathbf{f}_n \circ \mathbf{x}_n)$ for $\mathbf{x}_n \in b$
5:     Update $\boldsymbol{\theta}$ of IN by max $\mathcal{L}_L$ in Eq. 7.
6: **end for**
7: **return** Importance network (IN) parameters $\boldsymbol{\theta}$ which outputs local explanations $\boldsymbol{\alpha}_n$ for the input $\mathbf{x}_n$ in test-mini-batch.

---

### 4.3 Optimization

During the optimization, computing the gradient of Eq. 4 and 7 with respect to $\boldsymbol{\phi}_l$ requires obtaining the gradients of the integral (the first term) and also the KL divergence term (the second term), as both depend on the value of $\boldsymbol{\phi}_l$. The second term, the KL divergence between two Dirichlet distributions can be written in closed form,

$$\text{D}_{kl}[q(_l|\boldsymbol{\phi}_l)||p(_l|\boldsymbol{\alpha}_0)] = \log\Gamma(\sum_{j=1}^{S_l}\boldsymbol{\phi}_{l,j}) - \log\Gamma(S_l\alpha_0) - \sum_{j=1}^{S_l}\log\Gamma(\boldsymbol{\phi}_{l,j}) + S_l\log\Gamma(\alpha_0) + \sum_{j=1}^{S_l}(\boldsymbol{\phi}_{l,j} - \alpha_0)\left[\psi(\boldsymbol{\phi}_j) - \psi(\sum_{j=1}^{S_l}\boldsymbol{\phi}_{l,j})\right]$$

where $\boldsymbol{\phi}_{l,j}$ denotes the $j$th element of vector $\boldsymbol{\phi}_l$, $\Gamma$ is the Gamma function and $\psi$ is the digamma function. However, computing the gradient of the first term in Eq. 4 and 7 is not that straightforward. As described in Figurnov et al. (2018), the usual reparameterization trick, i.e., replacing a probability distribution with an equivalent parameterization of it by using a deterministic and differentiable transformation of some fixed base distribution[4], is one option. For instance, in an attempt to find a reparameterization, one could adopt the representation of a $k$-dimensional Dirichlet random variable, $\mathbf{f}_l \sim \text{Dir}(\mathbf{f}_l|\boldsymbol{\phi}_l)$, as a weighted sum of Gamma random variables, $\mathbf{f}_{l,j} = y_j/(\sum_{j'=1}^{K} y_{j'}), y_j \sim \text{Gam}(\boldsymbol{\phi}_{l,j}, 1) = y_j^{(\boldsymbol{\phi}_{l,j}-1)}\exp(-y_j)/\Gamma(\boldsymbol{\phi}_{l,j})$, where the shape parameter of Gamma is $\boldsymbol{\phi}_{l,j}$ and the scale parameter is 1. However, this does not allow us to detach the randomness from the parameters as the parameter still appears in the Gamma distribution, hence one needs to sample from the posterior every time the variational parameters are updated, which is costly and time-consuming.

Existing methods suggest either explicitly or *implicitly* computing the gradients of the inverse CDF of the Gamma distribution during training to decrease the variance of the gradients (e.g., Knowles (2015a), Figurnov et al. (2018), and Jankowiak & Obermeyer (2018)).

**Sampling.** In both objective functions, Eq. 1 and Eq. 7, in the left-hand side term (so called, cross-entropy term) we need to evaluate an integral over $\mathbf{f}$ (or $\mathbf{f}_n$). We do so by the Monte Carlo integration using Eq. 3 for the global setting and Eq. 6 for the local setting. The integral is evaluated for each data input $\mathbf{x}_k$ (we use

---

[4]For instance, a Normal distribution for $z$ with parameters of mean $\mu$ and variance $\sigma^2$ can be written equivalently as $z = \mu + \sigma\epsilon$ using a fixed base distribution $\epsilon \sim \mathcal{N}(0, 1)$.

here the $k$ data index to describe both the global and the local case):

$$\int q(\mathbf{f}_k|\boldsymbol{\alpha}_k)\log p_g(\mathcal{D}|\mathbf{f}_k)d\mathbf{f}_k \approx \frac{1}{J}\sum_{j=1}^{J}\log p(\mathcal{D}|\mathbf{f}_{k,(j)}), \tag{8}$$

where the subscript $(j)$ denotes the $j$th Monte Carlo sample $\mathbf{f}_k$ from the Dirichlet distribution $\boldsymbol{\alpha}_k$. Following (Knowles, 2015b), we compute the gradients of the integral implicitly using the inverse CDF of the Gamma distribution.

**A point estimate.** A computationally cheap approximation to the integral is using the analytic mean expression of the Dirichlet random variables,

$$\int q(\mathbf{f})\log p(\mathcal{D}|\mathbf{f})d\mathbf{f}|_{\mathbf{f}=\bar{\mathbf{f}}} \approx \log p(\mathcal{D}|\bar{\mathbf{f}}), \tag{9}$$

where $\bar{\mathbf{f}} = \frac{\boldsymbol{\alpha}}{\sum_{d=1}^{S}\boldsymbol{\alpha}^{[d]}}$. Computing the point estimate does not require sampling and propagating gradients through the samples, which significantly reduces the run time. In our experiments, we use both approximations where the specifics on each approximation are included in the Supplementary material.

### 4.4 Divergence for measuring similarity under BIF

We are interested in measuring how similar feature importance is under two different models. We denote the two Dirichlet distributions for feature importance obtained under the two models by $p$ and $q$, respectively, where $p$'s parameters are $\boldsymbol{\alpha} = [\alpha^{[1]}, \cdots, \alpha^{[S]}]$ and $q$'s are $\boldsymbol{\beta} = [\beta^{[1]}, \cdots, \beta^{[S]}]$.

The BIF's output, the Dirichlet distribution is an exponential family distribution, which we can write in terms of an inner product between the sufficient statistic $T(\mathbf{f})$ and the natural parameter $\boldsymbol{\eta}$:

$$p(\mathbf{f}|\boldsymbol{\alpha}) = h(\mathbf{f})\exp\left[\langle\boldsymbol{\eta}(\boldsymbol{\alpha}), T(\mathbf{f})\rangle - A(\boldsymbol{\eta})\right] \tag{10}$$

where $A(\boldsymbol{\eta}) = \log\int h(\mathbf{f})\exp(\langle\boldsymbol{\eta}(\boldsymbol{\alpha}), T(\mathbf{f})\rangle)d\mathbf{f}$ is the log-partition function, and $h(\mathbf{f})$ is the base measure. In case of the Dirichlet distribution, the natural parameter equals the parameter $\boldsymbol{\eta}(\boldsymbol{\alpha}) = \boldsymbol{\alpha}$, yielding a canonical form.

Under the exponential family distribution, popular divergence definitions such as the KL divergence $D_{KL}(p||q)$ and the Bregman divergence $B(q||p)$ can be expressed in terms of the log-partition function, its parameter, and the expected sufficient statistic:

$$\begin{aligned} D_{KL}(p||q) &= B(q||p) \\ &:= A(\boldsymbol{\eta}(\boldsymbol{\alpha})) - A(\boldsymbol{\eta}(\boldsymbol{\beta})) - \langle\boldsymbol{\alpha} - \boldsymbol{\beta}, \mathbb{E}_p[T(\mathbf{f})]\rangle, \end{aligned} \tag{11}$$

where $\mathbb{E}_p[T(\mathbf{f})]$ is the expected sufficient statistic under the distribution $p$. Under the Dirichlet distribution, all of the three terms are in closed-form, where the log-partition function is defined by $A(\boldsymbol{\eta}(\boldsymbol{\alpha})) = \log\Gamma(\alpha_0) - \sum_{d=1}^{S}\log\Gamma(\alpha^{[d]}), A(\boldsymbol{\eta}(\boldsymbol{\beta})) = \log\Gamma(\beta_0) - \sum_{d=1}^{S}\log\Gamma(\beta^{[d]})$, where $\Gamma$ denotes Gamma distribution, and each coordinate of the expected sufficient statistic is defined by $\mathbb{E}_p[T(f_d)] = \psi(\alpha^{[d]}) - \psi(\sum_{d}^{S}\alpha^{[d]})$, where $\psi$ is the digamma function. This allows us to evaluate the KL divergence conveniently. We demonstrate how we take advantage of having this easy-to-evaluate divergence in practice and examine the results under BIF in Sec. 5.3.

## 5 Experiments

We perform the experiments on both synthetic and real-world datasets. The binary synthetic datasets are meant to show the accuracy in selecting the appropriate features which were used to impact the label. The real-world datasets consist of both binary and multi-class datasets, including tabular and image data, and are meant to show broad applicability of the method. Finally, we present the need for well-tuned feature importance probabilities in privacy vs. explainability trade-off. In the experiments, we use the state-of-the-art benchmarks for comparison which allow for the instance-wise feature selection, that is L2X (Chen et al., 2018), INVASE (Yoon et al., 2018), SHAP (Lundberg & Lee, 2017a) and LIME (Ribeiro et al., 2016b).

## 5.1 Synthetic data

We first test our method on six synthetic datasets with the aim to identify the relevant features. We construct a data vector $\mathbf{x}$ in such a way that it is a random variable vector, $\boldsymbol{X} = [X^{[1]}, X^{[2]}, \ldots X^{[D]}]$, where the index describes an input feature. The first four binary synthetic datasets contain a fixed set of relevant features to test the global feature selection. Each data point consists of a 10-dimensional input feature $\boldsymbol{X} \sim \mathcal{N}(\mathbf{0}, \mathbf{I})$ and the associated label that depends on a subset of its features in such a way that $p(y = 1|\boldsymbol{X}) = \frac{1}{1+r}$ and $p(y = 0|\boldsymbol{X}) = \frac{r}{1+r}$ where the particular $r$ is defined by

- **Syn1a:** $\exp(X^{[1]}X^{[2]})$,
- **Syn1b:** $\exp(X^{[1]} - X^{[2]})$,
- **Syn2:** $\exp(\sum_3^6 (X^{[i]})^2 - 4)$,
- **Syn3:** $\exp(-100 \sin(2X^{[7]}) + 2|X^{[8]}| + X^{[9]} + \exp(-X^{[10]}))$.

In the remaining three datasets, we introduce an extra variable, $X^{[11]}$ which selects which set of features determine the label $y$, thus indirectly influencing the result, as well. As a result, a label $y$ depends on an alternating set of features which tests for local feature selection where a set of relevant features varies across a dataset:

- **Syn4:** if $X^{[11]} < 0$, sampled from Syn1a, else Syn2.
- **Syn5:** if $X^{[11]} < 0$, sample from Syn1a, else Syn3.
- **Syn6:** if $X^{[11]} < 0$, sampled from Syn2, else Syn3.

Notice that the features of the two datasets do not overlap and we can uniquely distinguish the features which generated a given sample. We generate 10,000 samples for each dataset, using 80% for training and the rest for testing. Note that in case of Syn1-3, these features are static (suitable for studying global feature importance), while in the case of Syn4-6, the features are alternating (suitable for studying instance-wise feature importance). We use the *Matthews correlation coefficient (MCC)* (Matthews, 1975) to assess how well each method identified the relevant features.

Table 1 summarizes how each of the algorithms copes to uncover the ground truth important features in the synthetic datasets. We include three variations of the proposed method, global selection described in Sec. 4.1 and two ways to compute local explanations described in Sec. 4.2. The first one evaluates the full integral through sampling (which we denote by samp), while the second one uses a point estimate (which we denote by pe) computed analytically as a mean of the Dirichlet distribution parameters. In all of these variants, we first pre-train a model $g$, and then feed it to our framework where we freeze the model parameters, and only optimize the importance parameters. The proposed method excels particularly in the local setting, on a more challenging datasets, where in all three datasets outperforms the existing methods by a substantial margin, 10-20 percentage points.

|  | Syn 1a | Syn 1b | Syn 2 | Syn 3 | Syn 4 | Syn 5 | Syn 6 |
|---|---|---|---|---|---|---|---|
| BIF (global) | **100** | **100** | **100** | 85.0 | - | - | - |
| BIF (inst, samp) | **100** | **100** | **100** | 93.6 | **90.1** | **86.0** | 85.2 |
| BIF (inst, pe) | **100** | **100** | **100** | 82.9 | 84.0 | 79.0 | **85.6** |
| L2X | **100** | **100** | **100** | **95.1** | 66.0 | 64.5 | 73.8 |
| INVASE | **100** | **100** | **100** | 81.0 | 57.3 | 50.0 | 36.1 |
| SHAP | 98.8 | 99.6 | 98.9 | 93.2 | 59.1 | 59.0 | 49.3 |
| LIME | **100** | **100** | **100** | 27.9 | 29.8 | 34.5 | 19.5 |

**Table 1: Synthetic datasets** to detect ground truth features. The Syn1-3 datasets consists of a fixed set of globally invariant important features, while Syn 4-6 consists of varying sets of important features instance-wise. The average over 5 runs is reported. The higher MCC (Matthews correlation coefficient), the better. In BIF, (*inst, samp*) means instance-wise explanation with sampling, while (*inst, pe*) means that with point estimate.

**Uncertainty.**   In the experiment featuring synthetic datasets, we also verify how well BIF can estimate the uncertainty of the importance values. While non-probabilitic methods provide only the point estimate, Bayesian approach allows to estimate how confident the algorithm is about its output, in this case, the feature importance value. Thus, Fig. 2 shows the posterior mean and variance of the importance vector. The posterior variance stands for the *confidence* the BIF algorithm has about the learned feature importance. Fig. 2 shows the variance of two datasets, Syn2 which is easier and Syn3 which is harder to predict its label. BIF consequently indicates the higher uncertainty for the harder dataset and lower for the easier dataset. This information can be particularly helpful when assessing the confidence about the importance of each feature, for example, it answers the question how probable it is that a given feature is the most important.

**Discussion.**   In selecting feature selection method, it is worth considering their advantages and disadvantages. The BIF global is a method whose number of parameters is linear in the number of features, however it works only in the global setting. The local variants require an additional network which may however work better, also in the global setting (by averaging the importance over all the data points). In terms of performance, the sampling and point estimate BIF produce similar results, however the shortcoming of the point estimate is that it produces the results with relatively high variance (see Supplementary materials for summary results). On the other hand, sampling is more time-consuming due to evaluations required for each sample. It is also worth discussing the effect of features on the label in synthetic datasets. In particular, Syn1a and Syn2 datasets affect the label positively, while in Syn1b there is a negative term, and in Syn3 the sine term may affect the label negatively. In both cases, the features have been detected successfully to large extent. Moreover, if we wish to know whether a feature affected the label positively or negatively, we may apply leave-one-out method, and run the method with and without the given feature.

## 5.2   Real-world data

**Tabular Data**   We consider *credit* (Cre) (license: DbCL v1.0) and *adult* (Dua & Graff, 2017) datasets with tabular input features and binary labels, and *intrusion* (Int) dataset with multi-class labels. Adult dataset predicts whether income exceeds some income threshold a year based on census data, credit dataset classifies applicants for credit availability, and the intrusion dataset classifies several types of burglaries.

This experiment consists of two parts. In the first part, we look for globally important features in the entire dataset. In the second part, we perform local feature search. As there is no known ground truth about the features, we evaluate the effectiveness of each method by selecting top $k$ features which are deemed most significant by a given method, and then perform the post-hoc classification task given only these $k$ input features (while removing the rest of the features). In the global setting, $k$ features are fixed for the entire dataset, while in the local setting each sample can select a different set of $k$ features. In the experiments, we standardize each feature to have a zero mean to mitigate the issue of out-of-distribution examples which could occur in the case of non-zero mean real-world features.

BIF outputs a probability distribution which directly allows to identify top $k$ features. On the other hand, INVASE and L2X output binary decisions for feature importance. For global explanations, we average the

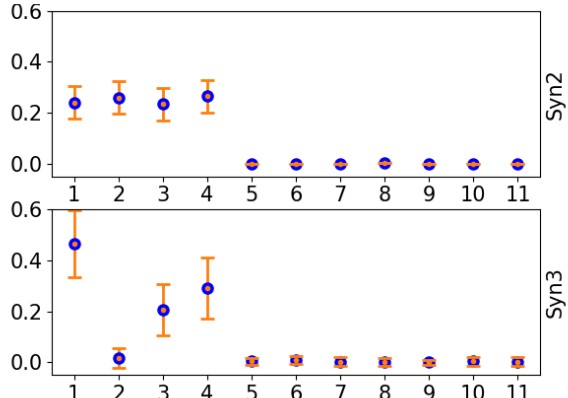

**Figure 2:** The illustration of the importance values and uncertainty learnt by BIF for the two datasets with four important values, Syn2 (Top) and Syn3 (Bottom). The algorithm is appropriately less certain about Syn3 ($\sum \sigma = 0.49$) which consists of features of varied importance (also less accurate) compared to Syn2 ($\sum \sigma = 0.25$).

|  | $k$ | Adult | | | Credit | | | Intrusion | | |
|---|---|---|---|---|---|---|---|---|---|---|
|  | | 1 | 3 | 5 | 1 | 3 | 5 | 1 | 3 | 5 |
| Local | **BIF** | **80.0** | **81.8** | 82,3 | **90.5** | **92.9** | **94.1** | 81.6 | **95.8** | 83.6 |
| | L2X | 78.6 | 81.7 | **83.1** | 86.5 | 89.1 | 92.8 | **81.9** | 79.0 | 77.4 |
| | INVASE | 73.5 | 78.6 | 82.1 | 81.4 | 90.9 | 91.5 | 70.5 | 76.6 | 45.5 |
| | SHAP | 71.8 | 74.8 | 76.9 | 86.8 | 84.9 | 84.5 | 69.1 | 72.6 | 73.4 |
| | LIME | 77.6 | 78.9 | 80.6 | 85.2 | 87.3 | 94.3 | 78.0 | 89.5 | **83.8** |
| Global | **BIF** | **78.2** | 76.5 | 82.4 | **96.1** | **94.9** | 94.9 | **82.3** | 82.4 | 82.6 |
| | L2X | 65.5 | 77.2 | 80 | 82.6 | 92.2 | 95.2 | 39.3 | 59.9 | 81.1 |
| | INVASE | 65.5 | **82.3** | 82.4 | 95.5 | 90.7 | 94.6 | 44.3 | 82.3 | 82.3 |
| | SHAP | 76.6 | 79.7 | **83.1** | **96.1** | 94.3 | **96.4** | **82.3** | **87.1** | **87.1** |
| | LIME | 77.6 | 78.9 | 75.9 | 92.4 | 88.9 | 92.5 | **82.3** | 81.3 | 83.4 |

**Table 2: Tabular datasets**. Classification accuracy as a function of $k$ selected features. **Up:** For gaining global explainabiilty. Same features are selected for all the datapoints. **Down**: For gaining local (instance-wise) explainability. A set of $k$ features is selected for each data point separately.

| | $k=1$ | $k=2$ | $k=3$ | $k=4$ | $k=5$ |
|---|---|---|---|---|---|
| **BIF** | **0.788** | **0.937** | **0.973** | **0.98** | **0.981** |
| **L2X** | 0.633 | 0.761 | 0.84 | 0.871 | 0.864 |
| **INVASE** | 0.584 | 0.78 | 0.901 | 0.915 | 0.905 |

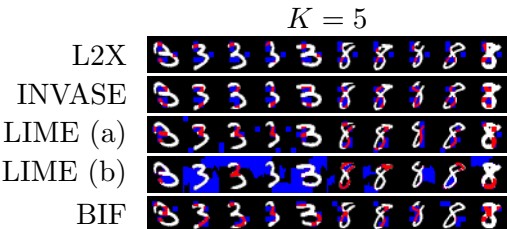

**Figure 3:** Quantitative and qualitative performance of BIF and the corresponding benchmarks on the MNIST image dataset. (Left) Post-hoc accuracy of MNIST classifier distinguishing digits 3 and 8 based on $k$ number of (4x4) selected patches. BIF outperforms other methods. (Right) Qualitative comparison of feature selection methods for 10 randomly selected instances of MNIST digits. The selected patches are highlighted in red and blue color. We show results for LIME both with our pre-determined segmentation into 4x4 patches (a) and using its own segmentation of the image pixels (b). BIF seems to particularly well include the differentiating curves between the digits 3 and 8 (see first, third, and fifth digit 8).

output for all data points, thus creating global ranking of features in INVASE and L2X. For local explanations, in case of L2X we can specify $k$ relevant features. INVASE has no such option and thus, for the fairest comparison, we use the selection probability given by the selector network as a proxy of importance score. In the global case of LIME, we average the rankings for the individual instances.

To summarize the results shown in Table 2 (classification accuracy averaged over five independent runs), we note that BIF performs well in both tasks, with a bigger edge in global search. We find that local explanation search is significantly more challenging than the global one, reflected in the lower classification accuracy. And so although the results provide good insight into which features are important locally, one should proceed with caution when relying on a subset of local features, especially in more risk-averse applications.

**MNIST Data.** Following (Chen et al., 2018), we construct a dataset with two labels by gathering the 3 and 8 digit samples from MNIST (LeCun et al., 2010) (license: CC BY-SA 3.0). We then train BIF, as well as L2X and INVASE models to select 4x4 pixel patches as relevant features. As the inputs have a dimensionality of 28x28, there are 49 features to choose from. To evaluate the quality of the selection, we first mask the test set by setting all non-selected patches to 0 and then use a classifier which was trained on unmasked data to compute the *post-hoc* accuracy on this modified test set. The post-hoc accuracies averaged over 5 runs are shown in Fig. 3a for different numbers of $k$ selected features.

The selection method differs between models. For BIF, we select the $k$ most highly weighted patches and for L2X, $k$ is set in advance. INVASE is treated differently, as the number of selected features varies and can

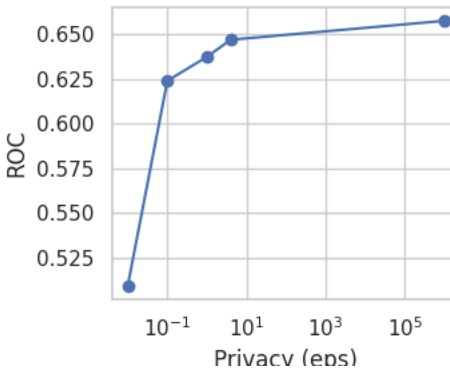 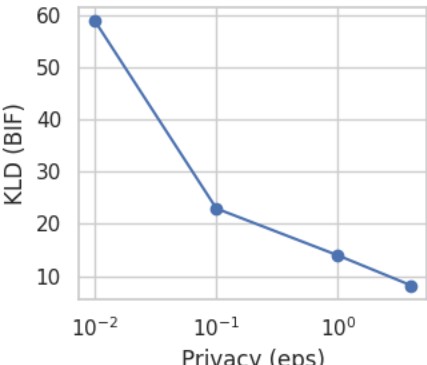

**Figure 4:** Privacy vs feature importance on Diabetes Readmission data (Rizvi et al., 2014). The x-axis indicates the privacy level of a classifier (smaller $\epsilon$ means more privacy). **Left:** The classification accuracy (ROC) improves as the privacy level decreases. **Right:** KL divergence between the feature importance distribution under the non-private classifier and that under the private classifier at the level that x-axis indicates. The feature importance learned by BIF at a stronger privacy (small $\epsilon$) has a larger divergence from the feature importance learned non-privately.

only be modified implicitly through the strength of the regularizer. So we tune the regularizer strength $\lambda$ to different values such that the average number of selected features equals $k$. The $\lambda$ values we use are 100, 50, 23, 18.5 and 15.5 for $k = 1, ..., 5$. We also show the qualitative results of each method in Fig. 3b.

### 5.3 Divergence for comparing feature importance distributions.

As described in Sec. 4, our method outputs the parameters of the Dirichlet distribution over the feature importance. With the deliberate choice of Dirichlet distribution, we can obtain a closed-form distance metric such as the KL divergence between two BIF's learned Dirichlet distributions. We exploit this to study trade-offs between important notions such as explainability, privacy, and fairness. In particular, we apply the KL-divergence to describe the level of explainability sacrificed at the cost of increase in privacy of a classifier. We show a similar experiment for the fairness trade-off in the supplementary materials.

We use the Diabetes Readmision dataset[5] (Rizvi et al., 2014) (license: CC0 1.0) to train a private classifier. We consider a private classifier using the differentially private stochastic gradient descent (DP-SGD) technique (Abadi et al., 2016), which perturbs the gradients during training to yield a classifier that guarantees a certain level of privacy, that is, it ensures that we cannot recreate the data that the model has been trained on. Fig. 4 (Left) shows how the classifier loses the accuracy measured in terms of the area under the curve as we increase the privacy level. Different privacy levels introduce different levels of noise we induced to the gradients during training (the higher $\epsilon$, the smaller the noise level, and $\epsilon = \infty$ corresponds to the non-private classifier). As the loss of accuracy is a known phenomenon, we aim to show a different effect, namely the impact of noise on the possibility to explain the data in form of the difference between the feature distribution without the noise and that when the noise is present at different levels.

The experiment shows two things. Firstly, as Fig. 4 and the KL divergence chart of Fig. 6 demonstrate, the relative differences in importance between features decrease as we increase the levels of noise. BIF-tuned probabilities well reflect the intuition that as we increase privacy levels, the explainability in form of assessing the correct distribution of feature importance decreases (e.g. the disparity between the top two features is much smaller for the private setting when $\epsilon = 0.1$, than for the non-private setting when $\epsilon = \infty$). In the Supplementary material, we also include the analysis regarding the INVASE, which shows that this intuition may not be exactly reflected in how features are selected. Secondly as Fig. 6 shows, even though the relative differences may be obfuscated by the noise, even at high levels of privacy we may distinguish the most relevant features, showing robustness of the feature selection ranking to the noise.

---

[5]We followed the data pre-processing given in https://www.kaggle.com/victoralcimed/diabetes-readmission-through-logistic-regression

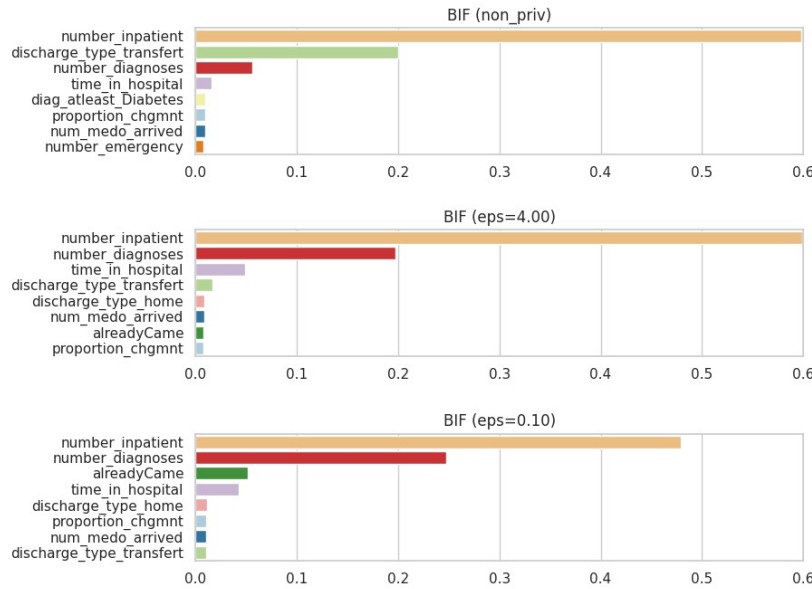

**Figure 5: BIF's learned feature importance**. We present the top 8 important features to the classifier trained with the Diabetes Readmission data (Rizvi et al., 2014) at a different level of privacy. Each feature is color-coded for better visualization. Smaller $\epsilon$ indicates higher levels of privacy. Relative differences between the features are affected by the increasing privacy (which is reflected in the decrease in KLD in the right Fig. 4.). The most important features in non-private setting remain important even for high levels of privacy, showing a level of robustness for identifying important features.

## 6 Conclusion

BIF proposes a comprehensive Bayesian framework for feature importance based on the Dirichlet distribution. The method works both in global and instanteneous settings, leveraging the benefits of neural networks. The Bayesian perspective allows to assess the confidence about the feature importance. The diverse results prove the method's robustness and the learnt distributions over features can be useful to measure explainability in a variety of applications.

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

## A   Appendix

You may include other additional sections here.

