# OpenReview forum: "Bayesian Importance of Features (BIF)"
_TMLR — Rejected by TMLR_

### Review · Reviewer_TfRZ · 2023-03-28

**Summary Of Contributions:**

This work proposes a new approach to quantify feature importance using a Dirichlet distribution. The idea is that each feature's importance score is a parameter in a Dirichlet distribution, and the distribution is fit via an objective that basically tries to preserve correct predictions after multiplying the input by a vector sampled from the distribution. This approach can be used in either a local or global fashion.

The experiments show encouraging results on synthetic and tabular datasets.

**Audience:**

Yes

**Broader Impact Concerns:**

No broader impact concerns.

**Claims And Evidence:**

No

**Requested Changes:**

Several experiment suggestions and writing clarifications are described above.

**Strengths And Weaknesses:**

### Strengths

The approach described in this paper is novel, and it's flexible enough to be applied to both local and global explainability. The local approach can be used in an amortized fashion using a learned network, which is interesting, and it involves a somewhat challenging problem of propagating gradients through the parameters of a Dirichlet distribution.

### Weaknesses

Representing feature importance via continuous Dirichlet parameters does not appear to make much sense. If you consider what this method does, it basically learns a distribution over probabilistic masks such that the model makes good predictions. But is it reasonable to focus on what the model does with arbitrarily downscaled feature values? These probabilistic masks create off-manifold inputs, and I'm not sure we should expect sensible outputs given 75% of one important feature and 25% of another, for example. And is it reasonable to force the mask to sum to 1 rather than some value $k < S$? In the case of high-dimensional data, like images, it seems like a major restriction to allow only $\approx 1$ pixel/patch to pass through the model.

Therefore, the premise of the paper seems to have a problem. I might have been convinced if the authors used high-dimensional datasets with a moderate number of important features, but the datasets shown here are mostly toy problems with a small number of important features.

I'll describe a couple other concerns below.

Method:
- This method is a lot like L2X, with the main differences being 1) it explicitly tries to output continuous masks rather than discrete ones, 2) the masks sum to 1 rather than a user-specified number $k$, and 3) it passes masked inputs through the original model rather than a separate one designed to accommodate missing feature values. These similarities should be discussed more explicitly, and the authors may want to dedicate more space to justifying these choices. According to my current understanding, these don't seem like improvements.
- The fact that the probabilistic masks create off-manifold inputs seems like an important issue in this method. I wonder if the authors could adjust the model somehow to work sensibly with downscaled inputs (I'm not sure how this would be possible), or if they could provide evidence that the model behaves sensibly without any modification. For example, I'm not sure we can make anything of a global mask that splits 25% probability between four important features; would the model even be able to make correct predictions after applying such a mask?
- Having a prior distribution over the Dirichlet parameters does not necessarily seem helpful. It's not clear how the authors set this prior, or whether the results would be meaningfully different if they simply turned it off. In finding which features are influential for the model, it's not clear why we should bias the results in any direction.
- Framing the ability to calculate explanation dissimilarity via KL divergence as an important contribution seems a bit misleading, because measuring explanation distance is even easier for non-probabilistic explanations: we can just calculate the Euclidean distance between feature importance scores.

Related work: there were many claims about related work that looked incorrect.
- In the introduction, the authors wrote that the literature currently proposes separate mechanisms for local and global explanations. That's not entirely true, there's a broad class of methods called removal-based explanations, of which L2X, INVASE, LIME and SHAP are members, that captures many local and global methods. These methods are all based on summarizing how the predictions change when features are withheld from the model [1]. In fact, it would be interesting for the authors to discuss the motivation for using continuous masks, given that discrete masks (as in L2X/INVASE) provide a way to actually remove feature information from the model, whereas continuous masks do not.
- Also in the introduction, the authors wrote that the current literature provides "two ways" of generating local explanations: model-agnostic methods and building models that provide explanations (which sounds like a description of L2X). This is a very incomplete description of the literature, there are many ways including inherently interpretable models, models with built-in interpretability mechanisms (e.g., self-attention or prototypes), gradient-based explanations, and removal-based explanations.
- Linking feature importance to feature selection by calling it a filter method doesn't sound correct. When we do feature attribution for images or text, for example, I'm not sure we should think of this as a filter method for feature selection.
- "Feature-additive methods" should probably just be called "feature attribution" or "feature importance" methods. The very first method the authors list, LIME, does not even satisfy the additivity property they refer to (the scores do not sum to the prediction minus the base rate prediction).
- SHAP is not a "weighted linear combination of features." It may be worth revisiting that work to describe it more accurately.
- What the authors currently call "attribution-based methods" (SmoothGrad, IntGrad) would probably be better referred to as "gradient-based methods." LIME and SHAP also generate feature attribution scores.
- Besides L2X, there's another method known as Real-X that exposed an information leakage issue in L2X/INVASE [2]. It would probably be worth mentioning this method and discussing whether the issue is present here or not.

[1] Covert et al, "Explaining by removing: A unified framework for model explanation" (2021)

[2] Jethani et al, "Have We Learned to Explain?: How Interpretability Methods Can Learn to Encode Predictions in their Interpretations"

Experiments:
- The authors did not specify how they set the KL divergence prior throughout the experiments.
- Similar to one of my comments from above: another ablation that would be worth including is the "point estimate" method with no KL divergence penalty, or the original method without the KL divergence penalty.
- When the authors ran LIME and SHAP, how did they handle held-out features? Given that the evaluations sometimes involve making predictions with most features set to zero, it seems sensible to run LIME and SHAP in that way. But I'm not sure this implementation choice was discussed.
- When generating global importance scores for LIME (and perhaps SHAP as well), the authors say they "average the rankings for individual instances." Calculating the mean absolute importance seems like a more reasonable choice here. Perhaps the authors could explain what they did and why in more detail.
- The MNIST 3 vs 8 dataset is a toy example. If the authors want to demonstrate that this method works with image data, it would be better to use a reasonably complex dataset with three color channels and higher resolution. It's hard to learn anything from the right side of Figure 3.

Presentation:
- "L2X outputs a predetermined set of important features" --> "L2X outputs a predetermined number of important features."
- The Figure 4 plots are cut off and shown in low resolution.

---

> ### Author Response · Authors · 2023-04-26
>
>
> ### Downscaling and off-manifold points
> The reviewer is right, we would not expect reasonable output when downscaling features. However, the purpose of the stage, when we learn the scores, is not obtaining good classification accuracy but assessing which of the inputs are more important for the model. In the post-hoc evaluation, the weighted values are not included, and no off-manifold points are created. The objective of the weights in the subsequent evaluation is twofold. Firstly, we may binarize the feature importance (according to some threshold) and given the importance information either keep a feature or remove it. Secondly, the weights allow us to assess the relative importance of one feature compared to the others. For example, if four features receive 25% score, it means they are equally important which indeed is the case for the synthetic dataset Syn2.
>
> ### KL-divergence vs Euclidean distance
>
> As we assume a probability distribution over features, KL-divergence which has a statistical interpretation is better suited when comparing or optimizing distributions.  KL-divergence appears in the method as a part of the derived ELBO (evidence lower bound) loss function (along with the likelihood term).  D<sub>KL</sub>(P|Q) describes in expectation how 'surprised' we are from using Q when the actual distribution is P. This is different than D<sub>KL</sub>(Q|P). This non-symmetric behavior is used when we derive ELBO and assume Q(w), that is distribution over features, is used instead of actual probability distribution P(w|x) where w are importance weights and x describes data [1,2]. On the other hand, Euclidean distance is a symmetric measure and misses this probabilistic interpretation. In sum, the advantage of the proposed Bayesian formulation is incorporating a better way to probabilistically compare features.
>
> At the same time, KL-divergence between prior distribution P(w) and the distribution we learn Q(w) provides a form of regularization where we do not allow the score values to be unbounded. Comparing feature importance via Euclidean distance could become problematic when the values are unbounded. Very large or very small scores may be the result of outliers, which can be expected especially data are scarce. BIF as a Bayesian method via the use of appropriate prior can mitigate this issue better than the existing methods.
>
> [1] Fox, C. and Roberts, S. A Tutorial on Variational Bayes.
> [2] MacKay D., Information theory, inference and learning algorithms
>
> ### No KL-divergence penalty
> As suggested by the reviewer, we perform a study where we include/exclude the KL-term. Interestingly, for the easier datasets (that is Syn1a, 1b and 2) inclusion of the KL-term does not make a difference and the method perfectly identifies the important features. However, for a harder dataset (Syn3) including KL-term improves the result, as without the KL-term the method more often misses one of the relevant features. Below are the numerical results (average over 5 runs). In the case of local datasets (Syn4-6), KL-term is even more desired since  it allows using higher learning rates (otherwise convergence error appears). In sum, the advantage of using KL-term is significant.
>
> |                    | KL   | no-KL |
> |--------------------|------|-------|
> | Syn1a                | 1.00 | 1.00  |
> | Syn1b           | 1.00 | 1.00  |
> | Syn2        | 1.00 | 1.00  |
> | Syn3 | 0.93 | 0.68  |
> | syn4               | 0.90 | 0.51  |
> | syn5               | 0.86 | 0.74  |
> | syn6               | 0.85 | 0.70  |
>
> ### Prior
>
> A priori, we do not know which features are more important than others. We, therefore, impose the uniform prior over the concentration parameters for the feature importance vector. We chose the particular value of alpha smaller than 1 to promote the sparse distribution such that most elements of the feature importance vector are close to zero while the majority of the probability mass is concentrated in a few elements of the vector.
>
> ### Held-out features
>
> In the case of held-out features, again to avoid off-manifold points, we take an average over the feature values and replace the feature values with the average.
>
> ### Literature
>
> We really appreciate the discussion and pointers about the feature selection methods and their categorization. It may be sometimes difficult to assess how methods relate to each other. We shall edit the literature review to provide a more diverse and complete categorization. Feature attribution and gradient-based methods are certainly better and more correct names. One class mentioned by the reviewer is also removal-based explanations which is a large class of methods that includes local and global methods, however, the methods are either local (SHAP, LIME) or global (Shapley value or SAGE[4]). We want to propose a framework that can be used both in global and local explainability.
>
> [4] Covert et al. Understanding Global Feature Contributions With Additive Importance Measures

---

> > ### Comment · Reviewer_TfRZ · 2023-05-04
> > **Response 1**
> >
> > Thanks to the authors for their response. I can comment on a couple points they raised and whether they fully addressed my concerns.
> >
> > ### Method design
> >
> > > The reviewer is right, we would not expect reasonable output when downscaling features. However, the purpose of the stage, when we learn the scores, is not obtaining good classification accuracy but assessing which of the inputs are more important for the model.
> >
> > This doesn't address my concern, because the method relies on downscaled features to assess which can preserve the model's original predictions. If such off-manifold inputs result in unreasonable model behavior, as the authors seem to acknowledge, why would we expect the method to learn anything useful via those off-manifold predictions? This question seems to require more careful consideration.
> >
> > > A priori, we do not know which features are more important than others. We, therefore, impose the uniform prior over the concentration parameters for the feature importance vector. We chose the particular value of alpha smaller than 1 to promote the sparse distribution such that most elements of the feature importance vector are close to zero while the majority of the probability mass is concentrated in a few elements of the vector.
> >
> > Setting a uniform prior seems sensible. But now that we've observed different results with different alpha values, this raises the question of how to choose the right alpha. The paper's synthetic experiments seem to permit tuning alpha to achieve good performance, but in general we don't have a ground truth to inform this hyperparameter choice. How exactly do the authors suggest setting the KL divergence penalty? The choice "smaller than 1" is a bit vague, does that mean 0.1 or 0.01? Should it depend on the dataset? It seems like the "right" choice depends on the scale of the loss from the predictions.
> >
> > > Feature selection, where we seek to extract a subset of features, is a field that inherently assumes a form of rivalry between features. This characteristic was also one of the motivations for the design of BIF where a fixed budget (here equal to 1) is given to features to split between them.
> >
> > Like I said in my review, it's not clear why a budget of 1 is a good choice, besides potentially simplifying computation. That budget seems somewhat arbitrary when there could be $k$ important features. In fact, it seems likely that this method would perform poorly when $k$ is moderate rather than very small.
> >
> > Relatedly, I just noticed that the authors required 4x4 superpixels when working with MNIST (an already low-resolution image dataset). As a result there are only 49 features, and it's conceivable that only a small number of superpixels matter. This seems to reflect my concern. I would be interested to know how this method performs with raw pixels, and also on a higher resolution image dataset (as mentioned in my review).
> >
> > Could the authors also clarify why BIF highlights some MNIST patches in blue and others in red? BIF doesn't output signed scores, so what do the colors mean? It looks like the patches are red when they overlap with the pen stroke and blue otherwise. If so, that's a bit strange and actually points to another downside of probabilistic importance scores: the scores don't reflect whether each feature is pushes the prediction up or down (which is an advantage of methods like SHAP, LIME and occlusion).

---

> > > ### Comment · Reviewer_TfRZ · 2023-05-04
> > > **Response 2**
> > >
> > > ### Literature
> > >
> > > > L2X also proposes only the solution for instancewise feature selection while BIF is a method that may be applied both in local and global selection. Proposing two procedures geared individually to both global and local selection allows for more suitable treatment of each approach.
> > >
> > > > One class mentioned by the reviewer is also removal-based explanations which is a large class of methods that includes local and global methods, however, the methods are either local (SHAP, LIME) or global (Shapley value or SAGE[4]). We want to propose a framework that can be used both in global and local explainability.
> > >
> > > In my opinion, this is overstating the generality of BIF relative to existing methods. If you consider L2X, a method that selects a specific number of features for a given instance, there's an analogous approach for the global setting called the Concrete Autoencoder [1]. (Note that although the method is designed for the unsupervised case, changing the objective is trivial.) INVASE has a couple of global analogues as well [2, 3]. These are typically used when training the original model, but they could also be applied in a post-hoc fashion. If that hasn't been done, I suspect it's due to concerns about off-manifold inputs.
> > >
> > > Similarly, SHAP and SAGE apply the same mechanism to local and global explanation, respectively. It's true that this Dirichlet masking trick can be applied in the local or global case, but other mechanisms can too, and I'm not sure this should be claimed as a point of novelty.
> > >
> > > [1] Balin et al, "Concrete autoencoders: Differentiable feature selection and reconstruction" (2019)
> > >
> > > [2] Yamada et al., "Feature selection using stochastic gates" (2020)
> > >
> > > [3] Chang et al., "Dropout feature ranking for deep learning models" (2017)

---

### Review · Reviewer_yn1U · 2023-04-11

**Summary Of Contributions:**

This paper presents two probabilistic algorithms which generate explanation for a pre-trained model's prediction at the global and local levels, respectively.

In both cases, the explanation comes in the form of a vector in a simplex -- i.e. non-zero components, all components sum up to one -- which indicates the relative importance of the input features.

For the global explanation algorithm:

The importance score vector is associated with the (learnable) parameter of a Dirichlet distribution over a simplex. The input to the pre-trained model can then be augmented via point-wise multiplication with samples drawn from such Dirichlet distribution. The induced prediction loss of the pre-trained model can then be used to generate a gradient signal to update the Dirichlet parameters.

For the local explanation algorithm:

The importance score vector is tailored specifically to each data point. This is achieved via a learnable neural net mapping from the original input to a customized parameter vector of a Dirichlet distribution, whose samples are used to augment input to the pre-trained model. The induced prediction loss of the pre-trained model can again be used to propagate gradient all the way back to the weights of the neural mapping (from local input to its corresponding Dirichlet parameter vector).

**Audience:**

Yes

**Claims And Evidence:**

No

**Requested Changes:**

Please consider address my points above in 1. to 7. Each of which has clear description on what needs to be changed.

**Strengths And Weaknesses:**

Strengths:

Overall, I find the idea interesting.
The writing is also clear, which makes the narrative easy to follow.
There is also a set of interesting experiments that show positive results

Weaknesses:

Although the idea & problem here are interesting, the proposed solution & its claimed contribution not well-demonstrated.
For example:

1. It is not clear which particular aspect of the proposed model is mainly responsible to the improved explainability of the proposed method (over previous work). Apparently, both the proposed approach and previous work such as L2X attempt to learn a score vector that characterizes the importance of each input feature. The difference is how each approach estimates such score vector but there is little discussion on the limitation of past solution and how the newly proposed method would address those.

2. The empirical studies are also inconsistent in terms of the adopted evaluation metrics. For the synthetic experiment, the MCC was reported but for other experiments, the classification accuracy is reported instead. The comparison and its drawn conclusion are therefore inappropriate & not holistic. I'd suggest using the same set of evaluation metrics across all experiments. I'd also recommend using previously established metric for explainability, e.g. those described in https://arxiv.org/abs/1802.01933

3. Furthermore, the experiment also lacks comparison with other approaches on many previously established benchmark. For instance, L2X was demonstrated well on a set of synthetic benchmark too. The authors should compare with L2X on those benchmarks in addition to any new benchmarks proposed in this paper.

4. The presented empirical results are also a bit confusing to me. L2X, LIME, SHAP are all approaches that generates (post-hoc) an score vector or a simple linear model that explains well the model's prediction at a particular data point. Thus, it is not clear what it means to compare those with the proposed approach under global explanation setting.

5. It is also not clear why we would want the importance score to sum up to one. That seems to create competition on the importance between inputs. Could the authors elaborate more on this?

6. The uncertainty calibration of the importance score is also barely demonstrated on two out of 6 synthetic datasets. For a comprehensive demonstration, the uncertainty calibration should have been reported across all datasets, including the real-world datasets. Please consider including this in the revised version.

7. More broadly, the motivation to adopt a probabilistic explaining model that has uncertainty calibration is somewhat weak. What is the practical use case for such uncertainty calibration?

---

> ### Author Response · Authors · 2023-04-26
>
> ### Metrics
>
> The reason the metrics are different is due to the different nature of each of the experiments. In the Experiment 1 we evaluate how well each method finds the important features. Given a set of features (some of which are relevant to the output label and some of which are not) we want to identify the relevant features. Since the datasets are synthetic we know the ground truth relevant features and can compare them to the ones identified by the method. Then the assessment involves comparing two sets for common elements, and so we found MCC (Mathew's correlation coefficient) to be most suitable metric for this task. Overall, this experiment aims to show that BIF is able to identify the relevant features.
>
> In the case of Experiment 2 we deal with real world datasets where we do not know the ground truth relevant features. The aim here is to reduce the feature space and retain the informative features relevant for the task. This can be shown by the classification accuracy of the model which is only given a subset of features (top-k features). If the features selected are irrelevant the accuracy should be low, and vice-versa. Here we show that BIF is able to detect relevant features in the noisy real-world datasets.
>
> Nevertheless, in Experiment 1, as the reviewer asked, we can use the same metric, classification accuracy as in Experiment 2. In the case of classification, we choose the features (as given in the Experiment 1) and test the model when the input that consists only of the selected features (exactly as it is done in Experiment 2). Then the results are presented below. We can note that the accuracy (almost) does not drop which shows the robustness of the features selected by BIF both in global and local settings.
>
> |       | global features | local features |all features |
> |-------|---------------|-------------|------------|
> | Syn1a | 0.97          | 0.97        | 0.97       |
> | Syn1b | 0.99          | 0.99        | 0.99       |
> | Syn2  | 0.96          | 0.96        | 0.96       |
> | Syn3  | 0.98          | 0.98        | 0.98       |
> | Syn4  | -             | 0.64        | 0.65       |
> | Syn5  | -             | 0.66        | 0.68       |
> | Syn6  | -             | 0.74        | 0.74       |
>
> ### Uncertainty estimation
> As suggested, we extend the experiments for estimating uncertainty. Below we show the numerical results for all the features for the synthetic datasets and adult real dataset (for space constraints we add credit and intrusion datasets into the paper appendix). For syn1-3, the metrics are addressed across all the data samples, in the case of syn 4-6 we select a random data sample. Interestingly, more important features have also higher uncertainty which also shows how well the method works for single examples (please refer also to the description in Sec.  how synthetic datasets are made).
>
> For a more practical use case, please consider the adult dataset which consists of 14 features which are meant to predict if an individual earns more or less than $50K. The first and fifth feature describe age and number of years of education, respectively. Both features are given the same importance (0.09 as presented in the second table below), however, the first feature has higher uncertainty (0.0052) than the fifth one (0.0037). We may conclude that although both age and education period are significant features to predict income, the method is more confident that education is a good predictor rather than age.
>
>
> | **var**|||||||||||||||
> | ----- | ------ | ------ | ------ | ------ | ------ | ------ | ------ | ------ | ------ | ------ | ------ | ------ | ------ | ------ |
> | syn1a | **0.0313** | **0.0313** | 0.0011 | 0.0009 | 0.0015 | 0.0013 | 0.0016 | 0.0006 | 0.0010 | 0.0008 | 0.0012 |        |        |        |
> | syn1b | **0.0212** | **0.0211** | 0.0015 | 0.0016 | 0.0020 | 0.0016 | 0.0011 | 0.0013 | 0.0017 | 0.0017 | 0.0018 |        |        |        |
> | syn2  | **0.0086** | **0.0083** | **0.0086** | **0.0087** | 0.0001 | 0.0001 | 0.0001 | 0.0001 | 0.0001 | 0.0001 | 0.0001 |        |        |        |
> | syn3  | **0.0370** | **0.0036** | **0.0175** | **0.0265** | 0.0013 | 0.0020 | 0.0017 | 0.0017 | 0.0017 | 0.0020 | 0.0013 |        |        |        |
> | syn4  | 0.0005 | 0.0006 | **0.0019** | **0.0026** | **0.0015** | **0.0020** | 0.0004 | 0.0000 | 0.0000 | 0.0000 |**0.0022**|        |        |        |
> | syn5  | **0.0087** | **0.0087** | 0.0004 | 0.0004 | 0.0001 | 0.0031 | 0.0015 | 0.0023 | 0.0003 | 0.0024 | **0.0105** |        |        |        |
> | syn6  | 0.0000 | 0.0000 | **0.0020** | **0.0009** | **0.0020** | **0.0020** | 0.0000 | 0.0000 | 0.0001 | 0.0000 | **0.0025** |        |        |        |
> | adult (mean) | **0.0052** (**0.09**) | 0.0017 (0.03)| 0.0025 (0.04)| 0.0024 (0.05)| **0.0037** (**0.09**) | **0.0070** (**0.17**) | 0.0020 (0.04) | 0.0024 (0.07) | 0.0033 (0.06) | 0.0043 (0.10) | 0.0040 (0.08) | 0.0027 (0.07)| 0.0037 (0.07)| 0.0018 (0.03) |

---

### Review · Reviewer_o8be · 2023-04-16

**Summary Of Contributions:**

The paper introduces two Bayesian methods for feature importance. The first is a global method that assumes a generative model that encodes the feature importance as a latent variable f generating the data distribution. For the local model, the authors make a similar assumption, except that each individual data point has its own set of parameters.

**Audience:**

Yes

**Broader Impact Concerns:**

The paper is about interpretability, which means there could be privacy concerns, but this is no more than concerns about any other existing interpretability methods.



**Claims And Evidence:**

Yes

**Requested Changes:**

I have added this in the weakness section.

please add attribution regarding the elbo bound. and if possible i would suggest adding more focus to the proposed method and reduce or move to appendix the buildup in sec 4.4 and evaluations in sec 5.3. Ideally i would have liked to see a more thorough investigation of the method, and extensions to say  influences of groups of data points to interpolate between global and local models.

**Strengths And Weaknesses:**

Strengths:

The paper is largely well-written. the proposed methodology is clearly communicated and motivated. the empirical evaluation is largely well done.  Experimental evaluations are interesting. the synthetic constructions show that the proposed method is consistently competent vs many state of the art methods. Real data experiments also show a similar trend.

Weaknesses:

In general integrating out over all f is intractable which is why the authors do an ELBO like lower bound. The way this is written comes across as if the authors came up with the jensen’s inequality and lower bound, I would suggest citing a reference regarding this.

Sec 4.3 suddenly uses \phi instead of f.

Sec 4.4 makes use of bregman divergences to calculate differences in importance features under two models. Since the Dirichlet distribution belongs to the exponential family, this calculation is easy (also known before). It is unclear from this section though why would be interested in comparing features across different models? It would be nice if the authors could add more motivation here explaining why this section is important and how it fits with the rest of the paper.

Similarly, Sec 5.3 makes use of this section i.e. section 4.4 to assess how relative feature importances change as more noise is added.  The results presented in this section are folk-lore and well-known/expected. I am not sure if they add anything of value to the literature in general, and definitely do not fit well with the rest of the paper which propose new Bayesian approaches for feature importances. I would have rather suggested the paper expands on those methods and introduce new ideas there. i understand the importance of using the special structure of the design in this work itself (use of dirichlet, which makes checking changes in feature importances easy due to easier evaluation of bregman divergences), but this could be done in other ways for other general models (e.g. a simple l2 distance evaluation).

---

> ### Author Response · Authors · 2023-04-26
>
> This is true that the results of the Sec. 5.3 (based on 4.4) are intuitive. However, we want to provide an example of being able to quantify and compare forms of explainability, a phenomenon that is still hardly defined in the literature. Comparing probability distributions under constraining regimes shows that BIF allows measuring well the loss in explainability when increasing the privacy (Sec. 5.3) or fairness (App. Sec. E) of the classifier. We should, however, consider moving both examples to Appendix as proposed by the reviewer, or perhaps the fairness example would be more interesting to the readers?
>
> In the released code, one may also train their own models for feature selection. From our observations, for models that converge, there is almost no difference in the feature importance that is output by BIF, which shows the method is fairly consistent across models.
>
> Please also refer to the Sec. Using KL-divergence vs Euclidean distance in the reply to Rev. 3 where we included some references regarding ELBO (which we also include in the paper).

---

### Author Response · Authors · 2023-04-26


### Thank you

Thank you to the reviewers for the detailed and exhaustive reviews. We appreciate that the method of the paper is seen as "novel", "well-motivated", and "flexible enough to be applied to both local and global explainability", the  paper is seen as "well-written", "clear" and "easy-to-follow",  and the experiments on both synthetic and real-world data are described to have "positive and encouraging results", and are "competitive vs many state of the art method"

Here we address a few more common points across the reviews, and subsequently, we include answers individually for each of the reviewers.

### Probability distribution

Feature selection, where we seek to extract a subset of features, is a field that inherently assumes a form of rivalry between features. This characteristic was also one of the motivations for the design of BIF where a fixed budget (here equal to 1) is given to features to split between them. In the process of training the importance weights, the network gradually phases out the less important features and pronounces the relevant features.

The related aspect is that since the feature importance belongs to the standard K-1 simplex, each feature importance is relative to the other feature’s values.  When features are treated independently, we may lose the ability to compare them. Through its interaction, the BIF importance scores can reach a sensible consensus. This is shown in the examples of synthetic dataset where BIF correctly identifies features as approximately equally important in case of Syn 1a or 2 datasets vs Syn 3 where the importance weights differ.

Besides, restricting the values between 0 and 1 is also a form of a soft generalization concerning the binary decision (which is hard to optimize in general) made in feature selection. Also, the optimization gets easier as we optimize over a real line (concentration parameters). Furthermore, framing the importance scores as probabilistic weights allows us to consider features in the context of synergies in the form of a weighted combination of features. As a final note, one may want to normalize the feature scores to one as a post-hoc activity. On the other hand, BIF through its probabilistic design naturally forms a distribution over features.

### Advantages in comparison to L2X

Although BIF and L2X may seem similar they are actually different under the surface. L2X is a method that derives from information theory and maximizes the mutual information between subsets of features and the response variables and approximates this quantity with a network that produces binary feature samples learned via continuous relaxation with the Gumbel-softmax trick. Using mutual information is widely appreciated in the feature selection literature, however as [3] indicates, when selecting relevant features MI obtains the higher number of false positives. Similarly, [4]  claims, mutual information may not always be adequate for feature selection (in case of regression). BIF, due to modeling the features via Dirichlet distribution, proposed an intuitive solution where we form a probability distribution over features, and thereby outputs continuous values which gives more insight in the importance of each feature.

L2X also proposes only the solution for instancewise feature selection while BIF is a method that may be applied both in local and global selection. Proposing two procedures geared individually to both global and local selection allows for more suitable treatment of each approach. In particular, in the case of global selection, we restrict the number of learnable parameters to the number of features, and therefore minimize the risk of overfitting. In the case of the local setting, we increase the expressivity of the method by including a neural network allowing for making decision for individual samples.

[3] Barraza et al. Mutual information and sensitivity analysis for feature selection in customer targeting: A comparative study

[4] Frénay et al. Is mutual information adequate for feature selection in regression?

### Local to global

As the reviewers indicated, the methods such as SHAP, LIME, L2X output the scores for each datapoint. To select globally important features we average the rankings across all the individual data points and select features with the highest mean rank. Averaging scores is also a solution, we tried both and the results for the given datasets were almost identical. Rankings remove some variations in scores and we found them to be slightly fairer solution for the benchmark methods.

---

### Decision · Action_Editors · 2023-05-19

**Recommendation:** Reject

**Comment:**

See the reviews and the discussion on "claims and evidence" above.

All reviewers agree that the method is potentially very interesting. However, in the current state, both the description of the novelty of the algorithm, and its experimental comparison to existing works, are not enough to understand whether it's really an improvement over existing methods, and if so, what is the key ingredient of this improvement. Thus, I will strongly encourage the authors to try to solve these issues, rewrite the paper and submit it again (either to TMLR or another venue).

I wish good luck to the authors with the next version of the paper.

**Audience:**

All researches interested in interpretable machine learning. Potentially a large audience.

**Claims And Evidence:**

The authors propose a new characterization of feature importance based on approximate Bayesian inference tools and Dirichlet distributions. The method is tested on synthetic and real data. These experiments seem to show an improvement over existing methods.

The paper is overall clear and easy to follow [Reviewers o8be and yn1U]. The experimental results are encouraging [o8be, yn1U, TrRZ].

However, the reviewers pointed out important problems in the paper:
1) insufficient or inexact comparison with the rich literature on feature importance, both on the methodology itself [Reviewer TfRZ pointed out wrong claims about existing methods] and in the experiments [yn1U, TfRZ].
2) insufficient experimental evidence, as the method is only tested on synthetic and *small* real data [TfRZ].

They also all pointed out minor problems that were essentially addressed by the authors during the discussion. However, aeven though the discussion was usefull, two reviewers still believe that the paper is not ready yet for publication [TfRZ: "I described several issues with the method that the authors weren't able to fully resolve. I also pointed to some issues with the experiments that I'm still waiting on the authors to resolve, and which seem to require adding new experiments to fully address"; yn1U: "Overall, the comparison with previous post-hoc interpretation work is insufficient & it is ultimately not clear which component of the proposed mechanism is responsible for the improvement (if any)" ... "I think this paper is not quite ready for publication yet."]

I agree with them, and will therefore recommend to reject the paper.